# In Vitro Analyses of Spinach-Derived Opioid Peptides, Rubiscolins: Receptor Selectivity and Intracellular Activities through G Protein- and β-Arrestin-Mediated Pathways

**DOI:** 10.3390/molecules26196079

**Published:** 2021-10-08

**Authors:** Yusuke Karasawa, Kanako Miyano, Hideaki Fujii, Takaaki Mizuguchi, Yui Kuroda, Miki Nonaka, Akane Komatsu, Kaori Ohshima, Masahiro Yamaguchi, Keisuke Yamaguchi, Masako Iseki, Yasuhito Uezono, Masakazu Hayashida

**Affiliations:** 1Department of Pain Medicine, Juntendo University Graduate School of Medicine, 2-1-1, Hongo, Bunkyo-ku, Tokyo 113-8421, Japan; yusuke.karasawa@viatris.com (Y.K.); y-kuroda@juntendo.ac.jp (Y.K.); a-komats@juntendo.ac.jp (A.K.); masahiro.yamaguchi@pfizer.com (M.Y.); keisuke@juntendo.ac.jp (K.Y.); miseki@juntendo.ac.jp (M.I.); hayashidatoday@yahoo.co.jp (M.H.); 2Department of Pain Control Research, The Jikei University School of Medicine, 3-25-8, Nishi-Shimbashi, Minato-ku, Tokyo 105-8461, Japan; minonaka@jikei.ac.jp (M.N.); 3b14025@alumni.tus.ac.jp (K.O.); 3Medical Affairs, Viatris Pharmaceuticals Japan Inc., 5-11-2, Toranomon, Minato-ku, Tokyo 105-0001, Japan; 4Division of Cancer Pathophysiology, National Cancer Center Research Institute, 5-1-1, Tsukiji, Chuo-ku, Tokyo 104-0045, Japan; kmiyano@ncc.go.jp; 5Laboratory of Medicinal Chemistry and Medicinal Research Laboratories, School of Pharmacy, Kitasato University, 5-9-1, Shirokane, Minato-ku, Tokyo 108-8641, Japan; fujiih@pharm.kitasato-u.ac.jp (H.F.); mizuguchit@pharm.kitasato-u.ac.jp (T.M.); 6Department of Anesthesiology and Pain Medicine, Faculty of Medicine, Juntendo University, 2-1-1, Hongo, Bunkyo-ku, Tokyo 113-8421, Japan; 7Medical Affairs, Pfizer Japan Inc., 3-22-7, Yoyogi, Shibuya-ku, Tokyo 151-0053, Japan

**Keywords:** analgesic, δ opioid receptor, G-protein-biased agonist, opioid peptide, rubiscolins

## Abstract

Activated opioid receptors transmit internal signals through two major pathways: the G-protein-mediated pathway, which exerts analgesia, and the β-arrestin-mediated pathway, which leads to unfavorable side effects. Hence, G-protein-biased opioid agonists are preferable as opioid analgesics. Rubiscolins, the spinach-derived naturally occurring opioid peptides, are selective δ opioid receptor agonists, and their p.o. administration exhibits antinociceptive effects. Although the potency and effect of rubiscolins as G-protein-biased molecules are partially confirmed, their in vitro profiles remain unclear. We, therefore, evaluated the properties of rubiscolins, in detail, through several analyses, including the CellKey^TM^ assay, cADDis^®^ cAMP assay, and PathHunter^®^ β-arrestin recruitment assay, using cells stably expressing µ, δ, κ, or µ/δ heteromer opioid receptors. In the CellKey^TM^ assay, rubiscolins showed selective agonistic effects for δ opioid receptor and little agonistic or antagonistic effects for µ and κ opioid receptors. Furthermore, rubiscolins were found to be G-protein-biased δ opioid receptor agonists based on the results obtained in cADDis^®^ cAMP and PathHunter^®^ β-arrestin recruitment assays. Finally, we found, for the first time, that they are also partially agonistic for the µ/δ dimers. In conclusion, rubiscolins could serve as attractive seeds, as δ opioid receptor-specific agonists, for the development of novel opioid analgesics with reduced side effects.

## 1. Introduction

Opioid analgesics are widely used as key medications for relief from pain, including perioperative pain, cancer pain, and nonmalignant chronic pain. However, their use is sometimes hampered in clinical practice owing to unfavorable side effects, such as tolerance, constipation, and respiratory depression [1,2]. Thus, the discovery of safer opioid analgesics is an urgent requirement. Opioid receptors (ORs), which belong to the G-protein-coupled receptor (GPCR) family [3], are classified into three subtypes—µ (MOR), δ (DOR), and κ (KOR)—and opioid analgesics mainly bind to MOR to exert their effects [4]. Internal signals from ORs are transmitted through two major pathways after the ligand conjugates with Gi/o proteins, followed by internalization of membrane receptors; the two pathways are the G-protein-mediated pathway that is required for analgesia, which is induced by decreasing the intracellular cAMP levels, and the β-arrestin-mediated pathway, which is associated with side effects [5,6]. Therefore, a biased analgesic with a pharmacological profile of favoring the activation of the G protein-mediated pathway over that of the β-arrestin-mediated pathway is desirable because it is considered to be effective and has fewer adverse events [7,8]. From this perspective, some molecules have been studied and indicated as G-protein-biased agonists in the past decades [9,10]. Among them, TRV130 (oliceridine) has been evaluated by intravenous administration in clinical studies and was approved as the first G-protein-biased agonist that can be used in clinical practice [11].

Besides MOR-selective agonists, there are several compounds selective for DOR or KOR that have been investigated in the preclinical studies [12,13]. They are expected to become alternatives for MOR agonists, which can cause severe side effects [14]. Compared with MOR agonists, DOR agonists show weaker effects in modulating acute nociception [12] but obvious effects in treating chronic pain under experimental conditions [15,16,17]. DOR can also be a therapeutic target for treating emotional disorders, such as depression [13,18]. However, none of the DOR agonists have been developed as an analgesic. Among the DOR agonistic compounds, rubiscolins are naturally occurring opioid peptides isolated from spinach leaves, produced by a pepsin digestion of d-ribulose-1,5-bisphosphate carboxylase/oxygenase (RuBisCO), the most abundant protein on earth [19,20]. Two types of rubiscolin—rubiscolin-5 and rubiscolin-6—exist, which are composed of penta- or hexa-amino acid residues (Tyr-Pro-Leu-Asp-Leu: YPLDL and Tyr-Pro-Leu-Asp-Leu-Phe: YPLDLF), respectively (Figure 1). Interestingly, these peptides showed antinociceptive effects upon p.o. administration in mice [21], which never occurs for endogenous opioid peptides. Moreover, rubiscolins are promising in terms of their unique effects other than analgesia, such as memory consolidation [22], anxiolytic effect [23], stimulation of food intake [24], enhancement of glucose uptake in skeletal muscle [25], and antidepressant-like effect [26]. Although the potency and actions of rubiscolins as G-protein-biased molecules were partially confirmed in a previous study using DOR [27], their in vitro profiles have not been sufficiently revealed.

The heterodimerization of ORs is also a noteworthy aspect [28,29]. It was recently revealed that ORs form heterodimers, which play an important role in pain modulation, and the selective ligand for the µ/δ opioid receptor (MOR/DOR) heteromer induced antinociception similar to that induced by morphine, but with less tolerance [30]. MOR/DOR heteromers have been reported to increase in cultured DRG neurons under pathophysiological conditions, such as chronic pain or subsequent exposure to morphine [31], and heterodimerization appears to be related to morphine-mediated antinociception and development of tolerance [32]. Therefore, MOR/DOR heteromers can also be targets for developing safer and more effective opioid analgesics [33,34]. We believe that it is preferable for opioid compounds to activate MOR/DOR heteromers, in addition to having a G-protein-biased property.

In the present study, we investigated the in vitro properties of rubiscolins in detail, including the agonistic or antagonistic effects for ORs and intracellular activities through the G-protein- and β-arrestin-mediated pathways, using MOR, DOR, KOR, and MOR/DOR heteromer.

## 2. Results

### 2.1. Effects of Rubiscolins on the Functions of ORs Evaluated Using the CellKey^TM^ System

The effects of rubiscolins on the three types of ORs (MOR, DOR, and KOR) were evaluated using the CellKey^TM^ system (MDS Sciex, Foster City, CA, USA) in HEK293 cells stably expressing Halo-tag^®^-MOR, T7-tag^®^-DOR, or Halotag^®^-KOR. Changes in cellular impedance were detected as activities of OR using this system. The changes in impedance induced by rubiscolins in positive controls of MOR (DAMGO), DOR (SNC-80), and KOR (U-50488H) were compared to confirm their agonistic effects on each OR. Rubiscolins showed dose-dependent effects only on DOR, whereas little effect was observed on MOR and KOR (Figure 2).

Moreover, we examined the antagonistic effects induced by a combination of rubiscolins with the positive control of MOR (DAMGO) or KOR (U-50488H), by comparing with the effect of a combination of each positive control with 10^−5^ concentration of a negative control for MOR (naloxone) or KOR (norbinaltorphimine: norBNI), respectively. Unlike for the combination with 10^−5^ concentration of negative control that completely suppressed the agonistic effects of the positive control for both MOR and KOR, rubiscolins had little antagonistic effects on MOR and KOR (Figure 3). These results suggest that rubiscolins act as selective DOR agonists without affecting the other subtypes (MOR and KOR) of ORs.

### 2.2. Effects of Rubiscolins on the Intracellular cAMP Levels Evaluated Using the cADDis^®^ cAMP Assay

The activities of compounds through the G-protein-mediated pathway were evaluated by measuring the intracellular cAMP levels for each OR using HEK293 cells stably expressing Halotag^®^-MOR, T7-tag^®^-DOR, or Halotag^®^-KOR (Figure 4). After obtaining results for rubiscolins, SNC-80 (a positive control for DOR), and KNT-127 (an existing selective DOR agonist used as a competitor) [35], we compared the effects of rubiscolins with those of other compounds, including each of the positive controls for the three types of ORs. The E_max_ and EC_50_ values (pEC_50_ defined as the negative logarithm of the EC_50_) for each OR were calculated (Table 1). As was observed for SNC-80 or KNT-127, rubiscolin-6 demonstrated a robust effect on DOR at 10^−5^ concentration. On the contrary, they had little effect on MOR and KOR; in contrast, KNT-127 showed full agonistic effects on both MOR and KOR. These results indicate that rubiscolins selectively activate the G-protein-mediated pathway of DOR to exert their pharmacological effects.

### 2.3. Effects of Rubiscolins on β-Arrestin Recruitment Measured Using the PathHunter^®^ Assay

To determine the activities of rubiscolins through the β-arrestin-mediated pathway, we performed the PathHunter^®^ β-arrestin recruitment assay using CHO-K1 cells stably expressing MOR and DOR (DiscoverX, Fremont, CA, USA), and U2OS cells stably expressing KOR (DiscoverX). We also evaluated the effects of SNC-80 and KNT-127. Compared with SNC-80, rubiscolins displayed little effect on DOR, as shown in Figure 5, whereas KNT-127 moderately recruited β-arrestin in DOR. Given these results, among the DOR-selective compounds used in the experiment, rubiscolin-5 and rubiscolin-6 were considered the most irrelevant with regard to the activity through the β-arrestin-mediated pathway. In contrast, all the DOR-selective compounds showed little effect on MOR and KOR, compared with each positive control.

### 2.4. Effects of Rubiscolins on the MOR/DOR Heteromer

Finally, we examined the effects of rubiscolins on the MOR/DOR heteromer through the G-protein-mediated pathway using the cADDis^®^ cAMP assay. As shown in Figure 6 and Table 2, rubiscolins acted as partial agonists, similarly to SNC-80, compared with ML335 [30], a specific agonist for MOR/DOR.

## 3. Discussion

In the present study, both rubiscolin-5 and -6 were indicated as G-protein-biased DOR full agonists without affecting MOR and KOR. The limited antagonistic effects of rubiscolins on MOR and KOR (Figure 3) confirmed using the three types of ORs, for the first time in this study, indicate that they rarely interfere with the cellular signaling mediated by endogenous or exogenous opioid ligands. The endogenous opioid system plays a critical role in modulating stress [36,37], anxiety [38,39], and the immune system [40]; hence, other than its role in analgesia, it is preferable that opioid agonists do not exert antagonistic effects on untargeted ORs, as these can lead to unexpected side effects that occur by attenuating the activities of endogenous ligands, such as enkephalins, β-endorphin, or dynorphin A. In addition, rubiscolins can potentially be administered in combination with exogenous ligands, such as MOR and KOR agonists and antagonists, without modulating their expected effects, which means that they are unique and attractive seeds that exhibit DOR selectivity, considering that the existing opioids can affect untargeted ORs to varying degrees [4]. As for the combination therapy of analgesics including opioids, opioid-sparing effects of non-opioid analgesics combined with opioids can reduce opioid consumption and its related side effects, especially in the perioperative pain management in terms of avoiding the toxicity and chronic use of opioids [41,42,43]. Rubiscolins can be novel candidates for use in combination with opioids. Therefore, further research is needed to investigate the efficacy and safety of DOR agonists, including rubiscolins, in combination with MOR agonists.

In view of the results of our cAMP assay, rubiscolins can be considered DOR-biased agonists, consistent with previous reports. However, compared with the findings in a previous study on the bias factor of rubiscolins, which found that rubiscolin-5 was more G-protein-biased than rubiscolin-6 [27], our results indicate that rubiscolin-6 is relatively stronger than rubiscolin-5 in activating the G-protein-mediated intracellular pathway, and the β-arrestin recruitment levels induced by rubiscolin-5 or rubiscolin-6 are equivalently negligible (Table 1 and Figure 5). Indeed, there is a structural difference between the two peptides, as rubiscolin-6 has Phe, an additional aromatic residue, at the sixth position. Although its function is not clear, rubiscloin-6 has been shown to have a higher receptor affinity and is about twice as potent in analgesia as rubiscolin-5 [21], as was observed in our study. Rubiscolin-6 has also been reported to have broad beneficial effects related to the central nervous system, other than analgesia [22,23,24,25,26]. As an example, for the development of DOR selective agonists, considering such effects, NC-2800 is under Phase 1 clinical study to determine the indication of major depressive disorder (https://jrct.niph.go.jp/en-latest-detail/jRCT2071210033 accessed on 30 September 2021). The development of DOR-selective agonists as alternative antidepressants is expected to offer a solution for the unmet need related to the patient’s adherence to the current treatment of depression, since their efficacy is independent of representative side effects of selective serotonin reuptake inhibitors, such as digestive symptoms [13]. Their antidepressant-like or anxiolytic-like activities are also desirable in the context of treating pain, considering psychological factors, such as depression and anxiety, are intimately associated with pain behavior, especially in chronic pain conditions [44,45,46]. Therefore, based on our results, we believe that rubiscolin-6 has more potential to be developed as a G-protein-biased DOR agonist than rubiscolin-5, not only as an analgesic but also as a medicine for treating other indications that are significantly different for DOR and MOR agonists, despite their mild analgesic properties compared with that of MOR agonists.

In OR signaling pathways, β-arrestin-mediated pathway is involved in unfavorable side effects, such as tolerance through the intracellular pathway in MOR or dysphoria through that of KOR. Interestingly, β-arrestin recruitment by rubiscolins on any type of OR was low, although moderate changes were observed even with KNT-127 for DOR (E_max_ (%): 35.4 ± 1.3), an existing selective DOR agonist [35], when compared with SNC-80 (E_max_ (%): 100.0 ± 2.6) (Figure 5B). Given these results, rubiscolins can be considered the safest among selective DOR agonists, possibly with fewer side effects, such as convulsion that sometimes occurs upon administration of DOR agonists [18], or increase in alcohol intake correlated with β-arrestin recruitment induced by DOR agonists [47].

Here, we report the effects of rubiscolins on MOR/DOR heteromers for the first time. Rubiscolins showed partial agonistic effects on the MOR/DOR heteromer (Figure 6C and Table 2). Moreover, the finding that rubiscolins have unique profiles in exerting their effects, mainly through the activation of the G-protein-mediated pathway in DOR, and in part through the MOR/DOR heteromer, is novel. In contrast, ML335 was reproduced as a full agonist of MOR/DOR, consistent with the findings in a previous study [30]. However, ML335 also acted as a full agonist for both MOR and DOR, and also partially recruited β-arrestin through both MOR and DOR. This suggests that there is still an unmet need to develop biased agonists that have more specific selectivity for MOR/DOR heteromers. A limitation of the present study is that we do not have data for the induction of β-arrestin recruitment on the MOR/DOR heteromer by rubiscolins, because it is not commercially available to investigate using the PathHunter^®^ β-arrestin assay. In addition, although little evidence has been obtained on how DOR-selective agonists affect the MOR/DOR heteromer, interestingly, recent research has suggested that simultaneous treatment with MOR agonists and DOR antagonists can modulate tolerance induced by MOR agonists [48,49]. Therefore, further research is required to decipher how rubiscolins act as G-protein-biased molecules for MOR/DOR heteromers, how they can contribute to analgesia and other effects, and to elucidate the utility of G-protein-biased MOR/DOR agonists.

Rubiscolins have the advantage of oral availability, although their absorption is not well known. In general, oligopeptides are thought to be metabolized by digestive enzymes (peptidases) and are then taken up in the form of dipeptides or tripeptides by the digestive organs, such as the stomach and small intestine. Whereas other opioid peptides are easily degraded, rubiscolins may not be disassembled and captured, and then pass through the blood–brain barrier to exhibit their antinociception [20]. They have been hypothesized to be resistant to proteolytic enzymes because of the Pro residue in the second position of their molecular structure, although it is not a characteristic feature of rubiscolins because the Tyr-Pro sequence at the N-terminus is generally present in the YP-type opioid peptides and is thought to be essential for opioid activity [21]. Thus, for the development of novel G-protein-biased DOR analgesics, further research is needed to decipher the mechanism of their uptake and to know whether some kind of active transporter is involved.

From a clinical perspective, the opioid crisis is currently a global challenge [50]. In general, opioid analgesics targeting MOR are shuffled to provide a “switching therapy”, so as to balance the benefits and risks of individual opioids [51]; sometimes, a rescue dose with immediate effect is added for breakthrough pain in cancer patients. However, the kinds of opioid analgesics are limited, and their dosages often tend to increase owing to the loss in efficacy, opioid-induced hyperalgesia, or tolerance. Therefore, analgesics with safer and more effective profiles with new mechanisms of action, such as rubiscolins, appear to be promising alternatives. Hopefully, they could be used for reducing the dosage of current opioids and for resolving the opioid crisis, as part of the opioid rotation strategy, if their efficacy is proven to be on par with that of the existing opioids.

## 4. Materials and Methods

### 4.1. Chemicals

The following reagents were used: D-Ala(2)-*N*-Me-Phe(4)-Gly-ol(5)-enkephalin (DAMGO), (+)-4-[(a*R*)-a-((2*S*,5*R*)-4-allyl-2,5-dimethyl-1-piperazinyl)-3-methoxybenzyl]-*N*,*N*-diethylbenzamide (SNC-80), trans-3,4-dichloro-*N*-methyl-*N*-(2-(1-pyrrolidinyl)-cyclohexyl)-benzeneacetamide (U-50488H), naloxone, norbinaltorphimine, forskolin, KNT-127, ML335 (Sigma-Aldrich, St. Louis, MO, USA); rubiscolin-5 (H-Tyr-Pro-Leu-Asp-Leu-OH) and rubiscolin-6 (H-Tyr-Pro-Leu-Asp-Leu-Phe-OH) were chemically synthesized by standard solid-phase peptide synthesis as described in Appendix A. Forskolin was diluted with dimethyl sulfoxide (DMSO) and other chemicals were diluted with water.

### 4.2. Cell Line

Human embryonic kidney 293 (HEK293) cells were obtained from American Type Culture Collection (ATCC^®^, Manassas, VA, USA), and HEK293 cells stably expressing Halotag^®^-MOR, T7-tag^®^-DOR, Halotag^®^-KOR, or Halotag^®^-MOR/T7-tag^®^-DOR were generated by transfection of the constructed plasmids using Lipofectamine reagent (Life Technologies, Carlsbad, CA, USA).

### 4.3. Cell Culture

HEK293 cells (stably expressing Halotag^®^-MOR, T7-tag^®^-DOR, Halotag^®^-KOR, or Halotag^®^-MOR/T7-tag^®^-DOR) were cultured in Dulbecco’s modified Eagle’s medium (DMEM) supplemented with 10% fetal bovine serum, 1% penicillin/streptomycin, and 5 μg/mL puromycin (InvivoGen, San Diego, CA, USA) for Halotag^®^-MOR, 250 μg/mL hygromycin B solution (FUJIFILM Wako Pure Chemical Corporation, Osaka, Japan) for T7-tag^®^-DOR, or 700 μg/mL genistein (Glico, Palo Alto, CA, USA) and 100 μg/mL hygromycin for Halotag^®^-KOR and Halotag^®^-MOR/T7-tag^®^-DOR. The incubation was done in a humidified atmosphere with 5% CO_2_ at 37 °C.

### 4.4. Functional Analysis of ORs Using the CellKey^TM^ System

The analysis was performed as described previously [52]. In brief, cells were seeded at a density of 5.0 × 10^4^ in CellKey^TM^ poly-d-Lysine (Sigma Aldrich, Saint Louis, MO, USA)-coated 96-well microplates with an embedded electrode at the bottom of each well and incubated for 24 h. After washing with CellKey^TM^ buffer composed of Hanks’ balanced salt solution (1.3 mM CaCl_2_∙2H_2_O, 0.81 mM MgSO_4_, 5.4 mM KCl, 0.44 mM KH_2_PO_4_, 4.2 mM NaHCO_3_, 136.9 mM NaCl, 0.34 mM Na_2_HPO_4_, and 5.6 mM d-glucose) containing 20 mM 4-(2-hydroxyethyl)-1-piperazineethanesulfonic acid (HEPES) and 0.1% bovine serum albumin (BSA), cells were incubated for 30 min at 28 °C, and then treated with vehicle or one of the reagents. The change in impedance of an induced extracellular current (dZiec) in each well was measured for 25 min, following a 5 min baseline measurement. The magnitude of change in the dZiec value was defined as ΔZiec, and the value for rubiscolins was calculated as a percentage using the highest value for each positive control.

### 4.5. Intracellular cAMP Assay with cADDis^®^

The assay was performed as described previously [53]. In brief, cells were seeded at 7.0 × 10^4^ cells/well on black-walled, clear flat-bottom 96-well plates with recombinant BacMam virus expressing the cADDis sensor and 0.6 μM sodium butyrate, and incubated for 24 h at 5% CO_2_ at 37 °C. The medium was replaced with 100 μL Krebs solution or pretreatment reagents. The 96-well plates were incubated at 28 °C for 30 min in the dark. Cell fluorescence was measured from the bottom of the plate using excitation/emission wavelengths of 485 and 525 nm, respectively, on FlexStation 3 (Molecular Devices, LLC., San Jose, CA, USA). Cells were stimulated with 50 μM forskolin to increase the cAMP levels. After 20 min, when the signal plateaued, cells were stimulated with the indicated drugs, and changes in fluorescence from each well were measured every 26 s for 40 min. Increase in fluorescence intensity reflects the decrease in cAMP, through the activation of Gi-coupled receptor. The data were transformed to changes in fluorescence over the initial fluorescence (ΔF/F_0_).

### 4.6. β-Arrestin Recruitment Assay with Pathhunter^®^

This was performed as described previously [54]. In brief, U2OS OPRM1, CHO-K1 OPRD1, or U2OS OPRK1 cells were seeded at a density of 1.0 × 10^4^ cells/well in 96-well clear-bottom white plates and incubated for 48 h. The cells were stimulated for 90 min (in the case of MOR and DOR) or 180 min (in the case of KOR) in a dilution series for each receptor at 37 °C under 5% CO_2_ and the PathHunter^®^ working detection solution was added. The luminescence intensity was measured using FlexStation 3 (BioTek Instruments Inc., Winooski, VT, USA) for 1 h at room temperature. Data are expressed as the maximum signal intensity of each test compound as a percentage of the maximum signal intensity of the positive control.

### 4.7. Statistical Analysis and Approval for the Study

Data are presented as means ± SEM for at least three independent experiments. Data from cADDis cAMP assays were analyzed using one-way ANOVA followed by Tukey’s multiple comparison tests. A value of *p* < 0.05 was considered statistically significant. All analyses and concentration–response curve fitting were performed using Prism 8 (GraphPad Software, San Diego, CA, USA). All experiments were approved and performed in accordance with the Guide for Genetic Modification Safety Committee, National Cancer Center, Japan.

## 5. Conclusions

In the present study, we showed that rubiscolins are G-protein-biased full agonists for DOR, as well as partial agonists for the MOR/DOR heteromers, with limited effects on endogenous ligands or opioid analgesics that activate MOR or KOR. Considering the evidence obtained, we believe that rubiscolins could serve as promising seeds for the development of novel, safer opioids and selective DOR agonists that can be orally used for treating pain.

## Figures and Tables

**Figure 1 molecules-26-06079-f001:**
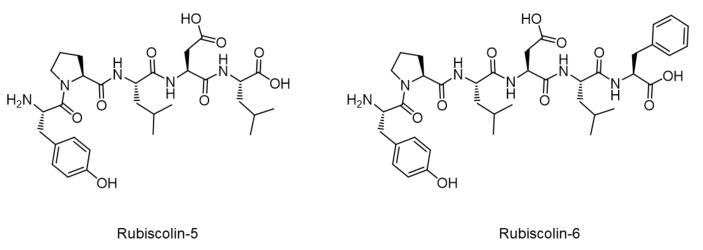
Molecular structures of rubiscolins.

**Figure 2 molecules-26-06079-f002:**
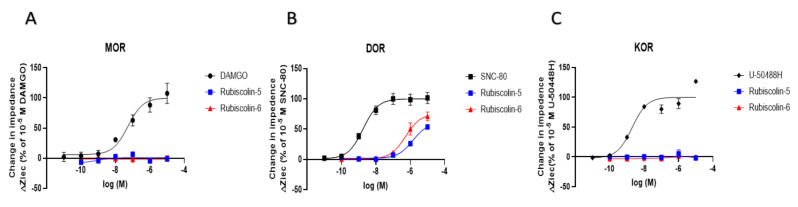
Effect of rubiscolins on MOR, DOR, and KOR, observed using the CellKey^TM^ system. The cells expressing MOR (**A**), DOR (**B**), and KOR (**C**) were treated with each compound (10^−11^–10^−5^ M), and changes in impedance (ΔZiec) were measured using the CellKey^TM^ system. Concentration–response curves were prepared by calculating ΔZiec relative to the data obtained for each positive control: 10^−5^ M DAMGO for MOR (**A**), 10^−5^ M SNC-80 for DOR (**B**), and 10^−5^ M U-50488H for KOR (**C**). All data points are presented as means ± S.E.M. for three independent experiments (*n* = 3–5).

**Figure 3 molecules-26-06079-f003:**
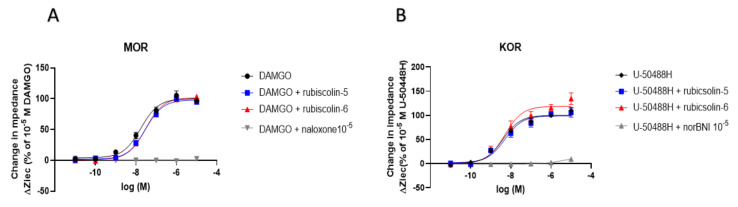
Evaluation of antagonistic effects induced by rubiscolins combined with positive control for MOR or KOR, observed using the CellKey^TM^ system. The cells expressing MOR (**A**) and KOR (**B**) were treated with each positive control alone or in combination with rubiscolin-5, rubiscolin-6, or 10^−5^ concentration of each negative control (10^−11^–10^−5^ M), and changes in impedance (ΔZiec) were measured using the CellKey^TM^ system. Concentration–response curves were prepared by calculating ΔZiec relative to the data obtained for each positive control: 10^−5^ M DAMGO for MOR (**A**) and 10^−5^ M U-50488H for KOR (**B**). All data points are presented as means ± S.E.M. for three independent experiments (*n* = 3–4).

**Figure 4 molecules-26-06079-f004:**
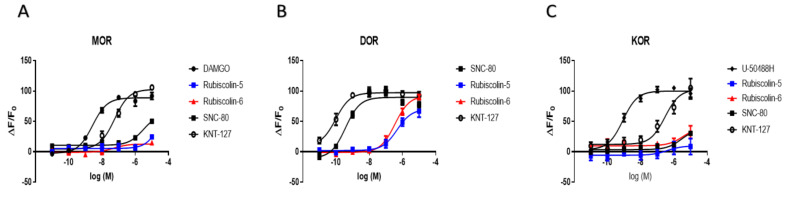
Changes in intracellular cAMP levels induced by rubiscolin-5, rubiscolin-6, and opioid compounds. Cells expressing MOR (**A**), DOR (**B**), or KOR (**C**) were treated with the listed compounds (10^−11^–10^−5^ M), and intracellular cAMP levels were measured with the cADDis**^®^** cAMP assay. Concentration–response curves were prepared by calculating cAMP levels relative to the data obtained with 10^−5^ M DAMGO for MOR (**A**), 10^−5^ M SNC-80 for DOR (**B**), and 10^−5^ M U-50488H for KOR (**C**). Data are presented as means ± S.E.M. for three independent experiments (*n* = 3–5).

**Figure 5 molecules-26-06079-f005:**
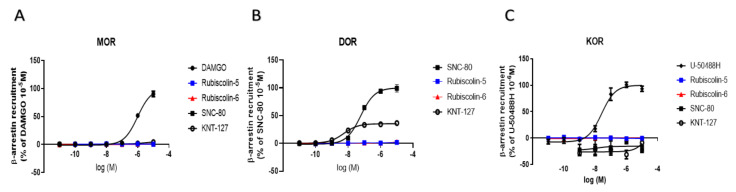
Levels of β-arrestin recruitment through OR induced by rubiscolin-5, rubiscolin-6, and opioid compounds. PathHunter^®^ β-arrestin assay was performed in cells expressing MOR (**A**), DOR (**B**), and KOR (**C**) by treating with each compound (10^−11^–10^−5^ M). Concentration–response curves were prepared by calculating intracellular β-arrestin levels relative to the data obtained for each positive control: 10^−5^ M DAMGO for MOR (**A**), 10^−5^ M SNC-80 for DOR (**B**), and 10^−6^ M of U-50488H for KOR (**C**). All data points are presented as means ± S.E.M. for three independent experiments (*n* = 3–6).

**Figure 6 molecules-26-06079-f006:**
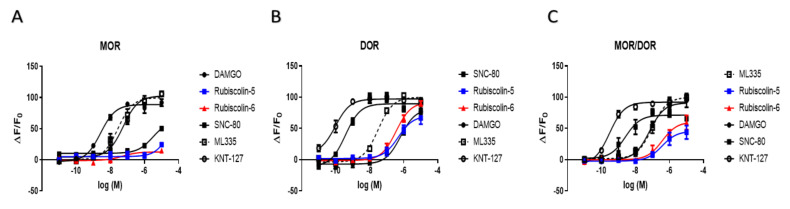
Changes in intracellular cAMP levels induced by rubiscolin-5, rubiscolin-6, and opioid compounds. Cells expressing MOR (**A**), DOR (**B**), or MOR/DOR (**C**) were treated with the listed compounds (10^−11^–10^−5^ M), and the intracellular cAMP levels were measured with the cADDis**^®^** cAMP assay. Concentration–response curves were prepared by calculating cAMP levels relative to the data obtained with 10^−5^ M DAMGO for MOR (**A**), 10^−5^ M SNC-80 for DOR (**B**), and 10^−5^ M ML335 for MOR/DOR (**C**). Data are presented as means ± S.E.M. for three independent experiments (*n* = 6–8).

**Table 1 molecules-26-06079-t001:** E_max_ and pEC_50_ values for rubiscolins and opioid compounds obtained in the cAMP assay for MOR, DOR, and KOR.

		MOR	DOR	KOR
E_max_ (%)	DAMGO	100.0 ± 3.0	90.9 ± 8.9	-
	SNC-80	69.8 ± 6.2 *	100.0 ± 3.6	37.7 ± 6.9 ^+^
	U-50488H	-	-	100.0 ± 3.0
	KNT-127	115.6 ± 4.3	108.4 ± 3.3	102.5 ± 9.5
	Rubiscolin-5	27.5 ± 5.5 *	78.4 ± 6.7 ^#^	9.6 ± 9.0 ^+^
	Rubiscolin-6	14.3 ± 3.2 *	103.0 ± 4.1	39.8 ± 20.5 ^+^
pEC_50_ (M)	DAMGO	8.5 ± 0.1	6.2 ± 0.2 ^#^	-
	SNC-80	5.5 ± 0.1 *	9.4 ± 0.1	5.6 ± 0.3 ^+^
	U-50488H	-	-	9.1 ± 0.1
	KNT-127	7.2 ± 0.1 *	10.0 ± 0.2	6.5 ± 0.2
	Rubiscolin-5	n.d.	6.3 ± 0.2 ^#^	n.d.
	Rubiscolin-6	n.d.	6.5 ± 0.1 ^#^	n.d.

E_max_ (means ± S.E.M.) and pEC_50_ (-LogEC_50_, means ± S.E.M.) were calculated according to the results shown in Figure 4. * *p* < 0.05 versus DAMGO, ^#^
*p* < 0.05 versus SNC-80, ^+^
*p* < 0.05 versus U-50488H. n.d.; not detected.

**Table 2 molecules-26-06079-t002:** E_max_ and pEC_50_ values for rubiscolins and opioid compounds obtained in the cAMP assay for MOR, DOR, and MOR/DOR.

		MOR	DOR	MOR/DOR
E_max_ (%)	DAMGO	100.0 ± 3.0	90.9 ± 8.9	91.1 ± 5.4
	SNC-80	69.8 ± 6.2 *	100.0 ± 3.6	71.4 ± 3.7 ^+^
	ML335	112.5 ± 5.3	111.6 ± 3.6	100.0 ± 4.2
	KNT-127	115.6 ± 4.3	108.4 ± 3.3	92.1 ± 2.4
	Rubiscolin-5	27.5 ± 5.5 *	78.4 ± 6.7 ^#^	45.8 ± 6.3 ^+^
	Rubiscolin-6	14.3 ± 3.2 *	103.0 ± 4.1	60.6 ± 6.3 ^+^
pEC_50_ (M)	DAMGO	8.5 ± 0.1	6.2 ± 0.2 ^#^	7.2 ± 0.2
	SNC-80	5.5 ± 0.1 *	9.4 ± 0.1	8.6 ± 0.2 ^+^
	ML335	7.6 ± 0.1 *	7.5 ± 0.1 ^#^	7.0 ± 0.1
	KNT-127	7.2 ± 0.1 *	10.0 ± 0.2	9.5 ± 0.1 ^+^
	Rubiscolin-5	n.d.	6.3 ± 0.2 ^#^	6.4 ± 0.3
	Rubiscolin-6	n.d.	6.5 ± 0.1 ^#^	6.4 ± 0.2

E_max_ (means ± S.E.M.) and pEC_50_ (-LogEC_50_, means ± S.E.M.) were calculated according to the results shown in Figure 6. * *p* < 0.05 versus DAMGO, ^#^
*p* < 0.05 versus SNC-80, ^+^
*p* < 0.05 versus ML335. n.d.; not detected.

## Data Availability

Not applicable.

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
