# Peer review of "In Vitro Analyses of Spinach-Derived Opioid Peptides, Rubiscolins: Receptor Selectivity and Intracellular Activities through G Protein- and β-Arrestin-Mediated Pathways"

_molecules, 2021, doi:10.3390/molecules26196079_

Round 1

Reviewer 1 Report

Karasawa and colleagues report pharmacological characterization of rubiscolin-5 and -6 peptides which have potential as analgesics by acting as selective agonists of delta opioid receptors. The authors employ several in vitro assays to dissect the activity of these compounds. The paper is well written and the experimental part is well documented and has been carefully carried out. My main concern is in regards to the novelty of these data – the selectivity for DOR and the signaling bias for the G-protein pathway has been previously reported (e.g. doi: 10.1016/j.euroneuro.2018.12.013). The only novel aspect is the activity at MOR/DOR heterodimers, but it is unclear whether the activity seen is due to binding at MOR/DOR heterodimers or DOR/DOR  and MOR/MOR homodimers that could form statistically equally well in this co-expression system? Is there any factor in this experiment that favors formation of the heterodimers ?

However, if this latter point is resolved and perhaps discussed in more detail I would reconsider a revised manuscript.

Minor points:

1) Table 1 and 2 gives logEC50 values, but from the data it is clear that these values are in the nM (10-9) or microM (10-6) range. Hence, the log values should be negative. Alternatively change log EC50 to  pEC50 (where pEC50 = -logEC50).

2) For all the figures F2-F6, it would be helpful to use the same symbols thoughout for control and test compounds.

3) Chemical synthesis details and HPLC and MS data should be included for the synthesis of rubiscolin-5 and -6.

4) It is my understanding that the ORs signal through Galphai resulting in reduced cAMP production. Why do the figures 4 and 6 show (or suggest) an increase in intracellular cAMP in response to ligand ?

Author Response

Reply to the Reviewers’ comments:

We thank the Reviewers for their careful review of our manuscript and for providing constructive comments. We have considered each comment in turn and have reviewed some articles quoted in the revised manuscript to improve our paper following the reviewers’ comments.

Our alterations are indicated in a point-by-point response to each reviewer’s comments as follows: listed below are the changes we made based on the reviewers’ feedback (indicated by italics).

Comments of Reviewer #1:

  1. Table 1 and 2 give logEC50 values, but from the data it is clear that these values are in the nM (10-9) or microM (10-6) range. Hence, the log values should be negative. Alternatively change log EC50 to  pEC50 (where pEC50 = -logEC50).

    Response: We thank the reviewer for the valuable comment. We added the definition of pEC50 as “(pEC50 defined as the negative logarithm of the EC50)” in Line 162 (Revised version), and then we changed every “log EC50 to “pEC50” described in Table 1 and 2.

  1. For all the figures F2-F6, it would be helpful to use the same symbols throughout for control and test compounds.

Response: We agree with the reviewer’s comment. As suggested, we have revised all the figures (Figure 2 to 6) by using the same symbol for each control and test compounds.

  1. Chemical synthesis details and HPLC and MS data should be included for the synthesis of rubiscolin-5 and -6.  

Response: We thank the reviewer for the valuable comment. We have revised the sentence regarding synthesis of rubiscolin-5 and -6, as “rubiscolin-5 (H-Tyr-Pro-Leu-Asp-Leu-OH) and rubiscolin-6 (H-Tyr-Pro-Leu-Asp-Leu-Phe-OH) were chemically synthesized by standard solid-phase peptide synthesis as described in supplemental data.” (Revised version; Page10, Line 338-340). We accordingly would like to submit the attached PDF file as supplementary material.

  1. It is my understanding that the ORs signal through Galphai resulting in reduced cAMP production. Why do the figures 4 and 6 show (or suggest) an increase in intracellular cAMP in response to ligand ?

Response: We thank the reviewer’s comment. As you pointed out, activation of Gi signal results in cAMP decrease. In Figure 4 and 6, we showed change of fluorescence intensity on the Y-axis. The cADDis assay for Gi decreases fluorescence intensity when cAMP is increasing in the cells and increases in fluorescence in response to activation of Gi (decrease of cAMP). We added the sentence as “((fluorescence increases reflect cAMP decreases through activation of Gi-coupled receptor)” in the method section to clarify that. (Revised version; Page 12, Line 388-389)

Reviewer 2 Report

This study confirmed and extended the previous findings by Cassell et al, Eur. Neuropsychopharmacol. 2019,29, (3), 450-456, indicating that naturally occurring peptides Rubiscolin-5 and −6 bind to δ opioid receptors and  do not recruit β-arrestin pathway. In the present paper Karasawa et al.  evaluated the properties of rubiscolins, in more detail, using CellKeyTM assay, cADDis® cAMP assay, and PathHunter® β-arrestin recruitment assay in cells stably expressing µ, δ, κ, or µ/δ heteromer opioid receptors. They found that rubiscolins are G protein-biased δ opioid receptor agonists with little agonistic or antagonistic effects for µ and κ opioid receptors, and that these peptides are  partially agonistic for the µ/δ heterodimers. The methods are sound and the manuscript has been clearly written, The obtained results are interesting and, in many aspects, quite original. Some limitations of this study, e.g., lack of data for the induction of β-arrestin recruitment on the MOR/DOR  heteromer by rubiscolins have been indicated in the Discussion.

Minor:

  • Discussion, Page 7: …for the first time in this sudy… (should be study).
  • Discussion, Page 8: "Given these results, rubiscolins can be considered the safest among selective DOR agonists, possibly with fewer side effects, such as convulsion that usually occurs  upon  administration  of  DOR  agonists [18]". The problem of   involvement of delta opioid receptor in seizure phenomena is still controversial. Therefore the word “usually” should be replaced by “sometimes”.

Author Response

Reply to the Reviewers’ comments:

We thank the Reviewers for their careful review of our manuscript and for providing constructive comments. We have considered each comment in turn and have reviewed some articles quoted in the revised manuscript to improve our paper following the reviewers’ comments.

Our alterations are indicated in a point-by-point response to each reviewer’s comments as follows: listed below are the changes we made based on the reviewers’ feedback (indicated by italics).

1.Discussion, Page 7: …for the first time in this sudy… (should be study).

Response: We thank reviewer comment. We revised the word accurately as “study”. (Revised version; Page 9, Line 233)

2. Discussion, Page 8: "Given these results, rubiscolins can be considered the safest among selective DOR agonists, possibly with fewer side effects, such as convulsion that usually occurs -upon  -administration  of  DOR  agonists [18]". The problem of   -involvement of delta opioid receptor in seizure phenomena is still controversial. Therefore, the word “usually” should be replaced by “sometimes”.

Response: We thank the reviewer for the valuable comment. We agree and have changed the word “usually” to “sometimes”. (Revised version; Page 10, Line 283).